# Hyperspectral Imaging and Machine Perfusion in Solid Organ Transplantation: Clinical Potentials of Combining Two Novel Technologies

**DOI:** 10.3390/jcm10173838

**Published:** 2021-08-27

**Authors:** Margot Fodor, Julia Hofmann, Lukas Lanser, Giorgi Otarashvili, Marlene Pühringer, Theresa Hautz, Robert Sucher, Stefan Schneeberger

**Affiliations:** 1Department of Visceral, Transplant and Thoracic Surgery, Medical University of Innsbruck, 6020 Innsbruck, Austria; margot.fodor@tirol-kliniken.at (M.F.); Julia.Hofmann@i-med.ac.at (J.H.); giorgi.otarashvili@i-med.ac.at (G.O.); Marlene.Puehringer-Stuhrmayr@i-med.ac.at (M.P.); Theresa.hautz-neunteufel@i-med.ac.at (T.H.); 2OrganLife, Organ Regeneration Center of Excellence, 6020 Innsbruck, Austria; 3Department of Internal Medicine II, Innsbruck Medical University, 6020 Innsbruck, Austria; Lukas.Lanser@i-med.ac.at; 4Department of Visceral, Transplant, Thoracic and Vascular Surgery, Leipzig University Clinic, 04103 Leipzig, Germany; Robert.Sucher@medizin.uni-leipzig.de

**Keywords:** organ transplantation, hyperspectral imaging, machine perfusion

## Abstract

Organ transplantation survival rates have continued to improve over the last decades, mostly due to reduction of mortality early after transplantation. The advancement of the field is facilitating a liberalization of the access to organ transplantation with more patients with higher risk profile being added to the waiting list. At the same time, the persisting organ shortage fosters strategies to rescue organs of marginal donors. In this regard, hypothermic and normothermic machine perfusion are recognized as one of the most important developments in the modern era. Owing to these developments, novel non-invasive tools for the assessment of organ quality are on the horizon. Hyperspectral imaging represents a potentially suitable method capable of evaluating tissue morphology and organ perfusion prior to transplantation. Considering the changing environment, we here discuss the hypothetical combination of organ machine perfusion and hyperspectral imaging as a prospective feasibility concept in organ transplantation.

## 1. Introduction

The organ preservation process has been an important focus of research over the last years. In the context of transplantation, this procedure is needed in order to maintain organ quality from the point at which organs are removed from the donor, during storage and transportation, and until transplantation into the recipient [1]. Preservation techniques include static cold storage (SCS) as gold standard in the past and, more recently, different dynamic machine perfusion (MP) techniques. Current approaches to organ preservation are categorized by: (1) method of delivery (static versus MP, continuous versus combination); (2) temperature (subzero, hypothermic, sub-normothermic, and normothermic); (3) oxygen (±nutrient) delivery; (4) location (ex situ versus in situ) [2]. A relevant aspect of MP is the possibility to test organ viability prior to transplantation. The normothermic system provides some advantages due to the physiological temperature and organ function inherent to this method [2]. Viability assessment may involve organ-specific injury markers, functional markers, or genetic markers collected during perfusion [2]. Due to the lack of an accurate predictive organ evaluation strategy, certain organs were rejected although they would still have been suitable for transplantation. The decision to accept or reject an organ for transplantation depends primarily on the expertise of the transplant team rather than on an objective measurement [3]. Therefore, optical evaluation tools to monitor organ quality before transplantation would be of great interest.

## 2. A Brief History and Overview of Machine Perfusion Techniques in Organ Transplantation

The era of SCS began in the 1960s, when Collins defined a method in order to perform kidney transport on ice using a preservation solution, with the result that organs could be conserved in a box after retrieval without damage after 30 h [4]. However, MP techniques were described some decades before. During the 20th century, where solid organ transplantation evolved to a clinical reality, a shift from the complicated technology of normothermic machine perfusion (NMP), previously described by Carrel in the 1930s [5,6], to a simpler technology proposing permanent availability and cost-effectiveness was aimed. Subsequently, in the 1960s, Belzer developed hypothermic machine perfusion (HMP) technology of kidney allografts, which gradually replaced NMP [7]. SCS allowed satisfactory results after solid organ transplantation, particularly for high-quality organs, with relative ease and low costs. However, in the light of an augmented usage of extended criteria donors (ECD) caused by the persistent organ shortage during the last decade, the use of SCS has been restricted. ECD liver grafts include categories such as advanced age, donation after circulatory death (DCD), hepatic steatosis, split liver grafts, cold ischemia time (CIT) >8–12 h, donors with increased risk of communicable disease transmission, those with active or past extrahepatic malignancy, hypernatraemia, and prolonged donor intensive care unit stay. ECD grafts are associated with an increased risk of early allograft dysfunction (EAD), delayed graft function (DGF), post-transplant complications, as well as higher rates for re-transplantations [8,9,10]. The key factor limiting the applicability of SCS in marginal grafts is given by the increased ischemia-reperfusion injury (IRI). In fact, during the hypoxic conditions of SCS, anaerobic metabolism leads to accumulation of toxic metabolites that contribute later to the manifestation of IRI the organ in the recipient [11,12]. The urgent necessity to improve the donor pool, as well as an ageing donor population with increasing co-morbidities, has resulted in an inevitable increase in using marginal organs [9,13]. Consequently, transplanting marginal organs in larger numbers required a reconsideration of alternative preservation methods, in order to preserve and eventually recondition marginal grafts [1]. In 2009, a landmark study concerning HMP in deceased donor kidneys was published, heralding a new era in organ preservation [14]. This trial showed that HMP reduced the incidence of DGF compared with SCS. Moreover, HMP allografts showed a lower risk of graft failure in the first year after transplantation [15]. Recently, a randomized multicentre trial showed that HMP leads to a decreased rate of non-anastomotic biliary strictures following the transplantation of livers obtained from DCD donors compared with conventional SCS [15]. NMP has the advantage of enabling a normal cellular metabolism, due to physiologic temperature. This technique allows a continuous perfusion with oxygenated blood or oxygen carrier, as well as medical and nutritional supplementation at body temperature during storage. This furthermore allows repairing of reversible injury and viability testing before transplantation through assessment of perfusion and biochemical parameters [1,9]. The first randomized trial confronting NMP and SCS was published in 2018 and represented a milestone in the field of liver transplantation [16]. This study allowed a first step in demonstrating that NMP is feasible, safe, and effective in clinical practice [16].

## 3. Spectral Cameras for Image-Guided Organ Transplantation

The digitization of pre-and intraoperatively imaging procedures offers the possibility of both the real-time analysis of acquired image data during the medical procedure and the visualization of the automatically extracted information for the surgeon [3]. In this way, space- and time-resolved imaging of clinical-relevant information, such as tissue type, or potential structures at risk are obtained during surgery. Hyperspectral imaging (HSI) is currently evaluated as a potential tool to expand the existing intraoperative imaging procedures with the aim of assessment of tissue types as precisely as possible [17]. Actually, different camera solutions with various acquisition systems are described in literature. Tunable filter spectrometers capture one wavelength at a given time. The system has to switch through the different channels to capture one spectral cube by combining multiple images. These images always have a temporal offset to each other. Push-broom spectrometers capture one spectral slice per image readout. To generate a hyperspectral cube, the device has to be scanned over the object. The hyperspectral snapshot imager captures x, y, and lambda in the same sensor read out. The data cube is taken at once without any timely offset [3]. HSI has been evaluated as an additional technique in order to measure tissue physiology, morphology, and composition [17]. Due to its capacity to capture ultraviolet and infrared wavelength spectra and based on the absorption/reflectance of the analyzed tissue, HSI acquires two-dimensional spatial images, ultimately formed into a three-dimensional data set called the hypercube [18]. This technique can be used to monitor the physiologically relevant tissue parameters, operating among other spectral ranges in the region of the optical window where the most relevant absorbers are melanin, lipids, water, and hemoglobin [3,17]. Subsequent imaging procedures generate pictures of the chemical tissue composition indicating the oxygen saturation (StO_2_), tissue perfusion (near-infrared perfusion index, NIR), tissue hemoglobin index (THI), and tissue water index (TWI) [19]. Previous studies demonstrated the utility of HSI in order to enhance the surgeon’s visualization beyond gross macroscopic assessment and may have impact on surgical guidance through tissue characterization [20]. Various clinical devices for the diagnostic support in a wide range of specialties are currently available [21].

## 4. Combination of Hyperspectral Imaging with Machine Perfusion: Is It Feasible?

In light of the emerging technology of MP, focusing on organ restoration and viability assessment before transplantation, the introduction of a non-invasive, simple, cost-effective additional tool, capable of measuring organ perfusion and eventually able to predict graft dysfunction or surgical complications, is still warranted. During solid organ transplantation, adequate delivery of oxygen at the tissue level is critical for maintaining graft viability and promoting aerobic metabolism [22]. Principally, different compartments within the single organs should be considered, in order to evaluate distinct functions. Observing the case of liver and kidney, the parenchymal, vascular, and immunological compartments were separated and suitable viability criteria assessed during MP were validated [23,24]. Currently utilized HSI parameters such as tissue oxygenation, tissue perfusion indices, and organ hemoglobin indices might be of subordinate importance if measured during SCS; however, in the era of MP, an HSI device integrated on an MP device as a fixed installation might provide useful data on organ viability and performance on the machine, integrating and potentially replacing serial expensive blood tests [25]. Evaluating different MP techniques, HMP does not represent the ideal setting to evaluate organ function, due to the low temperatures and severely reduced metabolic rate [26]. However, the statement that viability assessment is only feasible at normothermic temperatures should be applied with caution; in fact, while HMP was demonstrated to protect against mitochondrial injury [27,28], numerous proteins and metabolites are additionally released during hypothermic and sub-normothermic MP [29]. On the other hand, NMP offers more possibilities in terms of viability assessment. In fact, at normothermia and in the presence of oxygen, cellular metabolism resumes, and therefore assessing both organ injury as well as residual function can take place during NMP. Serial HSI measurements during MP might allow a quantitative evaluation of graft oxygenation and micro-perfusion, as well as organ hemoglobin and water concentration (Figure 1, Figure 2 and Figure 3). This should provide an additional information regarding organ quality before transplantation. 

Real-time monitoring of vascular flow and tissue oxygenation indicates viable tissue and facilitates early detection of impaired perfusion. In contrast to commonly used methods for the determination of the oxygenation status, HSI allows a pixel-wise analysis of chemical changes. The experimental results of HSI-based oxygen saturation calculation showed that HSI is suitable for the monitoring of the oxygen saturation distribution and the identification of areas with a reduced oxygen supply prior to transplantation [30]. Given the circumstance that HSI is able to graduate parenchymal damage, it could be helpful in observing eventual dynamic changes over MP, in evaluating their impact on post-operative adverse clinical conditions, such as DGF, EAD, or anastomosis site leakage, and in intervening before these manifestation occur. In fact, the majority of conventional clinical and histologic predicting tools either fall short of graft quality assessment, or require invasive and time-consuming evaluation of biopsies [31]. Additionally, the application of HSI during MP provides the surgeon of a dynamic “oxygenation map” of the whole organ, with no-touch visualization extended into the infrared and near-infrared wavelength regions. This allows for the detection of critical areas and offers additional information about the heterogeneity of oxygen supply during MP. 

Preclinical studies in porcine kidney grafts showed that using HSI during MP, by formulating an oxygenation map, could provide a precise differentiation between perfused and non-perfused tissue regions. Moreover, detection of arterial occlusions is possible [32]. This signifies that HSI should also be used to monitor and eventually optimate technical MP details, such as the position of cannulas. Concerning the potential of HSI in solid organ transplantation, a recent study on human kidney allografts demonstrated that oxygenation and micro-perfusion were significantly decreased in patients with DGF. Both parameters were proposed for assessment of graft viability and function immediately after transplantation [25].

## 5. Future Approaches, Technical Challenges and Limitations

The clear advantages of an eventual combination of the HSI technology with MP are the rapid applicability with reproducible results within ten minutes and its reliable information about the perfusion state of the organs. Furthermore, the organ can be monitored throughout the perfusion period, and a possible negative impact on posttransplant outcome may be anticipated before transplantation. Combining HSI with MP is technically feasible. However, the future challenge of integrating a real-time imaging procedure into a clinical MP setting might primarily regard technical features. An optimal acquisition distance and the automated use under sterile conditions should allow the best possible reproduction and interpretation of the spectral information. Data analysis based on HSI is limited by the low penetration depth of the propagating light in the tissue, depending on the attenuation coefficient of the analyzed tissue. Hence, tissue injuries in deeper regions cannot always be detected [30]. The apparent simplicity of HSI makes it attractive for clinical use during MP; however, a standardized organ-specific protocol is still required. The use of different markers implies several critical points, not only the type of marker and its clinical relevance, but also the critical threshold in order to predict adverse conditions and perform a risk estimation. Although HSI during MP appears promising and feasible, validation in clinical trials will be needed before establishing routine application.

## 6. Conclusions

Camera-based measurements such as HSI allowing a quick and contactless real-time perfusion imaging, founded on chemical tissue characteristics could be of great interest regarding pre-transplant viability assessment. Based on the statement, that ischemia has a relevant impact on graft injury, monitoring of oxygen supply might provide a potential marker for assessment and early identification of functional limitations, particularly in case of marginal organs [30]. Hence, HSI may contribute to ease the decision-making process whether an organ is suitable for transplantation or not, by providing additional valuable information on perfusion quality and graft viability [3,25,33,34,35].

## Figures and Tables

**Figure 1 jcm-10-03838-f001:**
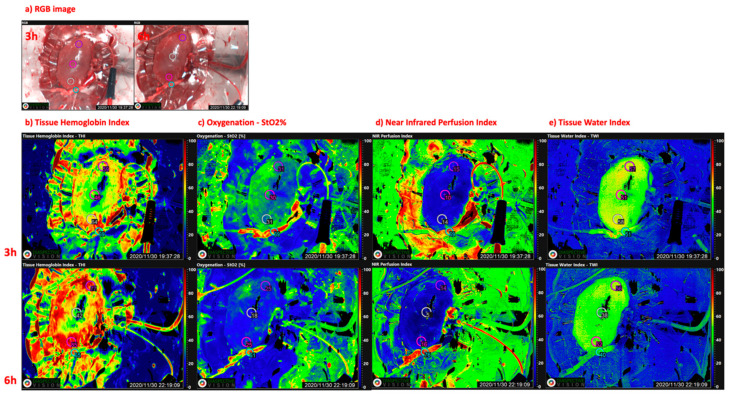
Hyperspectral imaging (HSI) applied on porcine kidneys after 3 h and 6 h NMP in an experimental setting; representative intraoperative (**a**) RBG image, corresponding HSI generated color coded, (**b**) tissue hemoglobin index (THI), (**c**) oxygen saturation index (StO2%), (**d**) near-infrared perfusion index (NIR), (**e**) tissue water index (TWI). Four standardized markers were placed on upper pole, corticomedullary boundary, lower pole, and ureter. Intraoperative images were acquired using the TIVITA^®^ Tissue HSI system (Diaspective Vision, Germany). For undisturbed image acquisition and data generation, the ambient light in the operating room had to be dimmed. The camera system incorporates a high number of spectrally differentiated channels, and acquires pictures with a high spectral resolution (5 nm) in the visible and near-infrared range (500–1000 nm). This scanner is mounted on a moving arm, which is brought to the organ. Using a 25 mm focal lens, a constant distance of 50 cm between the object and camera has to be retained during image acquisition. The HSI camera subsequently takes an RGB picture and in parallel computes a pseudo-color image, which represents physiologic parameters such as oxygen saturation, near-infrared perfusion, tissue hemoglobin, and tissue water of the recorded area. The maximum relative penetration depth of this HSI system is 6 mm. Quantitative StO2% measurements can either be performed at a depth of up to 1 mm for superficial microcirculation evaluation as well as at a depth of 4–6 mm, which corresponds to wavelengths recorded within the near-infrared spectrum. HSI can be applied almost “real time” and the acquisition for the hyperspectral image takes less than ten seconds. All images are stored, and further analysis of the hyperspectral data can be performed on the system with the TIVITA^®^ Suite software. HSI: hyperspectral imaging; NMP: normothermic machine perfusion; RGB: red-green-blue image; THI: tissue hemoglobin index; StO2%: oxygen saturation index; NIR: near infrared perfusion index; TWI: tissue water index.

**Figure 2 jcm-10-03838-f002:**
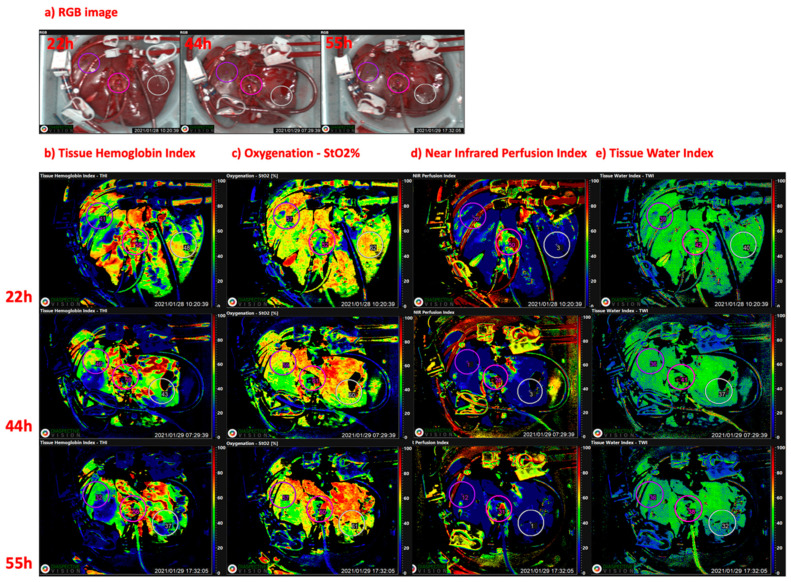
Hyperspectral imaging (HSI) applied on human livers after 22 h, 44 h, and 55 h NMP in an experimental setting; representative intraoperative (**a**) RBG image, corresponding HSI generated color coded, (**b**) tissue hemoglobin index (THI), (**c**) oxygen saturation index (StO2%), (**d**) near-infrared perfusion index (NIR), (**e**) tissue water index (TWI). Three standardized markers were placed on peripheral (right/left lobe) and central compartments. Intraoperative images were acquired using the TIVITA^®^ Tissue HSI system (Diaspective Vision, Germany). For undisturbed image acquisition and data generation, the ambient light in the operating room had to be dimmed. The camera system incorporates a high number of spectrally differentiated channels which acquires pictures with a high spectral resolution (5 nm) in the visible and near-infrared range (500–1000 nm). This scanner is mounted on a moving arm which is brought to the organ. Using a 25 mm focal lens, a constant distance of 50 cm between the object and camera has to be retained during image acquisition. The HSI camera subsequently takes an RGB picture and in parallel computes a pseudo-color image, which represents physiologic parameters such as oxygen saturation, near-infrared perfusion, tissue hemoglobin, and tissue water of the recorded area. The maximum relative penetration depth of this HSI system is 6 mm. Quantitative StO2% measurements can either be performed at a depth of up to 1 mm for superficial microcirculation evaluation as well as at a depth of 4–6 mm,, which corresponds to wavelengths recorded within the near-infrared spectrum. HSI can be applied almost “real time” and the acquisition for the hyperspectral image takes less than 10 s. All images are stored, and further analysis of the hyperspectral data can be performed on the system with the TIVITA^®^ Suite software. HSI: hyperspectral imaging; NMP: normothermic machine perfusion; RGB: red-green-blue image; THI: tissue hemoglobin index; StO2%: oxygen saturation index; NIR: near infrared perfusion index; TWI: tissue water index.

**Figure 3 jcm-10-03838-f003:**
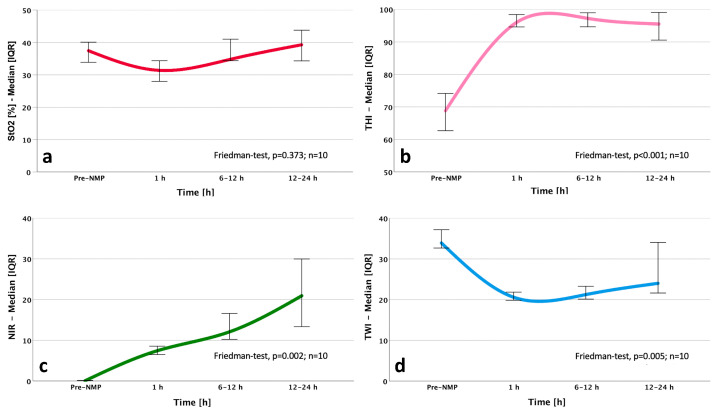
Courses of hyperspectral imaging (HSI) indices during normothermic machine perfusion (NMP) applied on human livers (**a**) Dynamics of tissue oxygen saturation index (StO2%), (**b**) tissue hemoglobin index (THI), (**c**) near-infrared perfusion index (NIR), and (**d**) tissue water index (TWI) during NMP of human livers (*n* = 10) in a clinical setting. NMP time was extended to a maximum of 24 h. HSI measurements were performed before starting NMP, after 1 h NMP, 6–12 h NMP, and 12–24 h NMP. In a retrospective analysis, markers representing the region of interest were inserted into the pseudo-colored images, and the index averages from the values inside the region of interest were calculated. Applied statistic test: Friedman Test, *p* values < 0.05 were considered significant. HSI: hyperspectral imaging; THI: tissue hemoglobin index; StO2%: oxygen saturation index; NIR: near infrared perfusion index; TWI: tissue water index; NMP: normothermic machine perfusion.

## Data Availability

All raw data on which this study is based will be made available by the corresponding author upon request.

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
