# Peer review of "Hyperspectral Imaging and Machine Perfusion in Solid Organ Transplantation: Clinical Potentials of Combining Two Novel Technologies"

_jcm, 2021, doi:10.3390/jcm10173838_

Round 1
Reviewer 1 Report
The article is labelled future perspective, and is a limited narrative review of the combined application of two technologies in organ transplantation. The text follows a simplified structure with sections for introduction, discussion and conclusions (the latter termed "limitations and future perspectives"). It includes illustrations of the use these techniques. The article concludes that their combined use is feasible but experimental studies are needed.
The concept of employing imaging or sensing technologies to examine the health of an organ harvested for donation is very intriguing and I find the article to be of interest to the scientific community.
However, there are a few issues I would like to address to improve the quality of the manuscript and make the text more accessible.
The text needs a revision to improve the English. Choice of words, phrasing and grammar can be improved.
Examples:
Line 38: “The era of SCS begun in the 1960s”, this should read “The era of SCS began in the 1960s"
Line 57: re-conditionate should read recondition
Line 82: combinate should read combine
Line 93: bi-dimensional and tridimensional are more commonly written two-dimensional and three-dimensional
Line 100: applicated on should read applied to
Line 151: are simply integrable in should read are easily integrated
The labelling of the paragraphs is unconventional. I think most readers would expect a final conclusions paragraph. There are two number two paragraphs (lines 37 and 78).
The introduction uses the terms high-risk organs and marginal organs but does not explain them in any way. The concepts should be introduced, since they are of great importance to the discussion on availability of organs, and the author should reflect on whether both terms are needed in the text.
The section labelled: “2. Combining hyperspectral imaging and machine perfusion; is it realistic?” begins to outline the utility of HSI to predict post-transplant organ failure and compares it to the use of more time-consuming biopsy techniques. While being common clinical practice, the use of biopsies does little to improve the quality of harvested organs. I think the author should revise the structure to focus more directly on the possibility of realtime feedback to improve the efficacy of perfusion techniques to improve the quality of the transplanted organs (such as in lines 115-117).
The technical challenges or requirements of setting up a system to combine machine perfusion and HIS are not addressed at all. I think they should be since a concluding remark is that it is feasible to combine HSI and machine perfusion. What is the processing time needed to evaluate hyperspectral images? Can it be automated for real-time use? What camera solutions are available or described in the literature?
The included figures need to be improved. The text within the figures is very small and a lot of zooming is needed to read the text. The legends are too sparse. Since there is no materials and methods-section there is a need for more elaborate texts to explain what is presented. The abbreviations (such as THI, tissue hemoglobin index) should be explained since some occur only in the figures, not in the text. The graph (fig 3) does not match the illustrations in fig 1 or 2 and thus need a separate explanation of the underlying premises. The relevance of the significance data in this figure is unclear. The trends are significantly correlated to ischemia time?
Author Response
Response to Reviewer 1 Comments
The article is labelled future perspective, and is a limited narrative review of the combined application of two technologies in organ transplantation. The text follows a simplified structure with sections for introduction, discussion and conclusions (the latter termed "limitations and future perspectives"). It includes illustrations of the use these techniques. The article concludes that their combined use is feasible but experimental studies are needed. The concept of employing imaging or sensing technologies to examine the health of an organ harvested for donation is very intriguing and I find the article to be of interest to the scientific community. However, there are a few issues I would like to address to improve the quality of the manuscript and make the text more accessible.
Point 1: The text needs a revision to improve the English. Choice of words, phrasing and grammar can be improved.
Examples:
Line 38: “The era of SCS begun in the 1960s”, this should read “The era of SCS began in the 1960s"
Line 57: re-conditionate should read recondition
Line 82: combinate should read combine
Line 93: bi-dimensional and tridimensional are more commonly written two-dimensional and three-dimensional
Line 100: applicated on should read applied to
Line 151: are simply integrable in should read are easily integrated
Response 1: Thank you for politely pointing this out. We edited the manuscript in order to improve the the choice of words, phrasing and grammar.
Point 2: The labelling of the paragraphs is unconventional. I think most readers would expect a final conclusions paragraph. There are two number two paragraphs (lines 37 and 78).
Response 2: Thank you for addressing this point. We revised the structure of the manuscript, the labelling of the chapters and included a conclusion paragraph.
Point 3: The introduction uses the terms high-risk organs and marginal organs but does not explain them in any way. The concepts should be introduced, since they are of great importance to the discussion on availability of organs, and the author should reflect on whether both terms are needed in the text.
Response 3: Thank you for this suggestion. We have introduced and explained the concept of marginal organs in the manuscript and also deleted the term “high-risk organ”, in order to clarify this issue.
Point 4: The section labelled: “2. Combining hyperspectral imaging and machine perfusion; is it realistic?” begins to outline the utility of HSI to predict post-transplant organ failure and compares it to the use of more time-consuming biopsy techniques. While being common clinical practice, the use of biopsies does little to improve the quality of harvested organs. I think the author should revise the structure to focus more directly on the possibility of realtime feedback to improve the efficacy of perfusion techniques to improve the quality of the transplanted organs (such as in lines 115-117).
Response 4: Thank you for addressing this. We revised the structure of the manuscript in some sections and focused more directly on the possibility of real-time feedback concerning perfusion quality assessment with HSI. Moreover, we mentioned additional advantages of this technique throughout the manuscript.
Point 5: The technical challenges or requirements of setting up a system to combine machine perfusion and HSI are not addressed at all. I think they should be since a concluding remark is that it is feasible to combine HSI and machine perfusion. What is the processing time needed to evaluate hyperspectral images? Can it be automated for real-time use? What camera solutions are available or described in the literature?
Response 5: Thank you for this suggestion. We have integrated these aspects in the manuscripts and re-structured the subchapters in order to achieve a better reading flow.
Point 6: The included figures need to be improved. The text within the figures is very small and a lot of zooming is needed to read the text. The legends are too sparse. Since there is no materials and methods-section there is a need for more elaborate texts to explain what is presented. The abbreviations (such as THI, tissue hemoglobin index) should be explained since some occur only in the figures, not in the text. The graph (fig 3) does not match the illustrations in fig 1 or 2 and thus need a separate explanation of the underlying premises. The relevance of the significance data in this figure is unclear. The trends are significantly correlated to ischemia time?
Response 6: Thank you for pointing this out. As you suggested, we improved the quality of the figures and elaborated the figure legends in order to clarify the proposed issues. The trends represented in Figure 3 are referred to the changing values of the HSI parameters over normothermic machine perfusion time which was maximal 24 hours.
Concerning the association of HSI parameters with the duration of cold ischemia time (CIT), we performed a subgroup analysis and looked for significant associations between the HSI indices and CIT £/> 360 minutes. We did not included these findings in the manuscript, because this was not a key issue of this paper, but we attached an additional table to this response.
Table 1: Association of cold ischemia time with HSI indices
|
HSI Index |
CIT <= 360 minutes |
CIT >360 minutes |
Total |
p-value # |
|
StO2 pre-NMP |
40 (35 - 48) |
32 (28 - 41) |
36 (31 - 43) |
0.020 |
|
StO2 1h |
31 (27 - 35) |
30 (23 - 38) |
31 (25 - 37) |
0.148 |
|
StO2 6-12h |
43 (34 - 43) |
38 (33 - 42) |
41 (34 - 43) |
0.437 |
|
StO2 12-24h |
44 (44 - 56) |
34 (33 - 41) |
39 (34 - 50) |
0.639 |
|
THI pre-NMP |
58 (52 - 74) |
73 (54 - 83) |
69 (52 - 77) |
0.057 |
|
THI 1h |
95 (92 - 95) |
99 (95 - 100) |
96 (93 - 99) |
0.464 |
|
THI 6-12h |
95 (92 - 99) |
99 (96 - 100) |
97 (95 - 99) |
0.437 |
|
THI 12-24h |
80 (80 - 96) |
99 (91 - 99) |
95 (80 - 99) |
0.073 |
|
NIR pre-NMP |
4 (0 - 8) |
0 (0 - 0) |
0 (0 - 4) |
0.238 |
|
NIR 1h |
8 (7 - 21) |
6 (3 - 8) |
7 (4 - 14) |
0.058 |
|
NIR 6-12h |
21 (10 - 32) |
15 (8 - 21) |
17 (10 - 22) |
0.053 |
|
NIR 12-24h |
24 (18 - 24) |
13 (0 - 31) |
21 (4 - 31) |
0.530 |
|
TWI pre-NMP |
33 (28 - 37) |
33 (29 - 36) |
33 (29 - 37) |
1.000 |
|
TWI 1h |
21 (20 - 24) |
19 (17 - 21) |
20 (19 - 22) |
0.219 |
|
TWI 6-12h |
23 (20 - 24) |
21 (19 - 24) |
21 (19 - 24) |
0.616 |
|
TWI 12-24h |
40 (26 - 41) |
22 (20 - 34) |
26 (22 - 40) |
0.030 |
Values are median (i.q.r.); # Mann-Whitney-U Test, p values<0.05 are considered significant
CIT, cold ischemia time; StO2, Oxygen saturation index; THI, Tissue hemoglobin index; NIR, Near infrared index; TWI, Tissue water index
Reviewer 2 Report
I would like to thank for the opportunity to revise this manuscript. This was a concise paper on the hyperspectral imaging applied to machine perfusion in the setting of liver transplantation. I really appreciate the manuscript and the topic, which can fall into the “perspective” section.
The paper is fluent and conclusions are, in my opinion, adequate.
I have only few comments:
- I suggest to add the recently published randomized trial on the NEJM about HMP in liver transplant (PMID 33626248)
- I agree that NMP allows a real-time evaluation of graft quality. Nevertheless, HMP seemed to protect against mitochondrial injury (as explained in PMID 30660724).
- Has this technique previously been applied in the setting of liver surgery?
- Can steatosis influence this technique?
- Has any other technique recently been proposed in this setting to evaluate ischemia/tissue perfusion?
Author Response
Response to Reviewer 2 Comments
I would like to thank for the opportunity to revise this manuscript. This was a concise paper on the hyperspectral imaging applied to machine perfusion in the setting of liver transplantation. I really appreciate the manuscript and the topic, which can fall into the “perspective” section.
The paper is fluent and conclusions are, in my opinion, adequate.
I have only few comments:
Point 1: I suggest to add the recently published randomized trial on the NEJM about HMP in liver transplant (PMID 33626248)

Response 1: Thank you for addressing this point. We have added this recently published trial in our manuscript.
Point 2: I agree that NMP allows a real-time evaluation of graft quality. Nevertheless, HMP seemed to protect against mitochondrial injury (as explained in PMID 30660724).
Response 2: Thank you for this suggestion. Although hypothermic technologies are frequently considered less helpful to assess liver viability, the same molecules found in NMP perfusate can also be identified during cold or sub-normothermic liver perfusion. We have added this reference in our manuscript.
Point 3: Has this technique previously been applied in the setting of liver surgery?
Response 3: This technique has been applied in the setting of anatomic left liver resection. A paper regarding this issue has been published in 2019 (“Hyperspectral Imaging (HSI) in anatomic left liver resection”, Sucher et.al) (PMC6731347). We added this reference in our manuscript.
Point 4: Can steatosis influence this technique?
Response 4: This is a very good point. Livers with significant levels of macrosteatosis may experience narrowing of the sinusoids with subsequent reduced perfusion flows. One consequence of sinusoidal obstruction due to fat droplets or to stuck blood cells is a reduced flow, which leads to secondary hypoxia and lower perfusion quality.
Hyperspectral imaging was previously applied for tissue classification (PMC6985687) and demonstrated to be able in differentiating also lipid tissue. Moreover, the recently developed Diaspective Vision devices (e.g. the TIVITA® Mobile device) can assess the fat content in the tissue area under consideration, by measuring the Tissue Lipid Index (TLI [index value]).
For this analysis, we used the TIVITA® Tissue Diaspective Vision device, which has not integrated the possibility to measure the tissue lipid content. To evaluate the impact of donor steatosis in our cohort, we performed a subgroup analysis (see Table 1 below). However, donor steatosis did not had impact on the single HSI indices over NMP perfusion time.
Table 1: Association of donor steatosis with HSI indices
|
HSI Index |
Absent donor steatosis |
Mild donor steatosis |
Moderate donor steatosis |
p-value # |
||||||
|
StO2 pre-NMP |
36 (34 - 40) |
33 (30 - 46) |
51 (51 - 51) |
36 (34 - 40) |
33 (30 - 46) |
51 (51 - 51) |
36 (34 - 40) |
33 (30 - 46) |
51 (51 - 51) |
0.429 |
|
StO2 1h |
33 (27 - 47) |
25 (24 - 37) |
28 (28 - 28) |
33 (27 - 47) |
25 (24 - 37) |
28 (28 - 28) |
33 (27 - 47) |
25 (24 - 37) |
28 (28 - 28) |
0.445 |
|
StO2 6-12h |
42 (31 - 43) |
41 (35 - 43) |
34 (34 - 34) |
42 (31 - 43) |
41 (35 - 43) |
34 (34 - 34) |
42 (31 - 43) |
41 (35 - 43) |
34 (34 - 34) |
0.684 |
|
StO2 12-24h |
39 (33 - 59) |
38 (38 - 41) |
44 (44 - 44) |
39 (33 - 59) |
38 (38 - 41) |
44 (44 - 44) |
39 (33 - 59) |
38 (38 - 41) |
44 (44 - 44) |
0.891 |
|
THI pre-NMP |
69 (51 - 74) |
74 (62 - 79) |
54 (54 - 54) |
69 (51 - 74) |
74 (62 - 79) |
54 (54 - 54) |
69 (51 - 74) |
74 (62 - 79) |
54 (54 - 54) |
0.725 |
|
THI 1h |
97 (95 - 99) |
95 (87 - 100) |
95 (95 - 95) |
97 (95 - 99) |
95 (87 - 100) |
95 (95 - 95) |
97 (95 - 99) |
95 (87 - 100) |
95 (95 - 95) |
0.749 |
|
THI 6-12h |
99 (96 - 100) |
97 (95 - 99) |
92 (92 - 92) |
99 (96 - 100) |
97 (95 - 99) |
92 (92 - 92) |
99 (96 - 100) |
97 (95 - 99) |
92 (92 - 92) |
0.310 |
|
THI 12-24h |
93 (80 - 97) |
99 (99 - 100) |
64 (64 - 64) |
93 (80 - 97) |
99 (99 - 100) |
64 (64 - 64) |
93 (80 - 97) |
99 (99 - 100) |
64 (64 - 64) |
0.094 |
|
NIR pre-NMP |
0 (0 - 2) |
0 (0 - 0) |
8 (8 - 8) |
0 (0 - 2) |
0 (0 - 0) |
8 (8 - 8) |
0 (0 - 2) |
0 (0 - 0) |
8 (8 - 8) |
0.209 |
|
NIR 1h |
8 (6 - 18) |
5 (1 - 8) |
19 (19 - 19) |
8 (6 - 18) |
5 (1 - 8) |
19 (19 - 19) |
8 (6 - 18) |
5 (1 - 8) |
19 (19 - 19) |
0.221 |
|
NIR 6-12h |
13 (10 - 18) |
20 (7 - 25) |
22 (22 - 22) |
13 (10 - 18) |
20 (7 - 25) |
22 (22 - 22) |
13 (10 - 18) |
20 (7 - 25) |
22 (22 - 22) |
0.557 |
|
NIR 12-24h |
27 (10 - 32) |
6 (0 - 13) |
24 (24 - 24) |
27 (10 - 32) |
6 (0 - 13) |
24 (24 - 24) |
27 (10 - 32) |
6 (0 - 13) |
24 (24 - 24) |
0.211 |
|
TWI pre-NMP |
37 (28 - 38) |
31 (30 - 34) |
22 (22 - 22) |
37 (28 - 38) |
31 (30 - 34) |
22 (22 - 22) |
37 (28 - 38) |
31 (30 - 34) |
22 (22 - 22) |
0.148 |
|
TWI 1h |
21 (19 - 24) |
20 (17 - 21) |
22 (22 - 22) |
21 (19 - 24) |
20 (17 - 21) |
22 (22 - 22) |
21 (19 - 24) |
20 (17 - 21) |
22 (22 - 22) |
0.553 |
|
TWI 6-12h |
22 (20 - 25) |
20 (18 - 24) |
23 (23 - 23) |
22 (20 - 25) |
20 (18 - 24) |
23 (23 - 23) |
22 (20 - 25) |
20 (18 - 24) |
23 (23 - 23) |
0.499 |
|
TWI 12-24h |
30 (21 - 40) |
23 (21 - 26) |
47 (47 - 47) |
30 (21 - 40) |
23 (21 - 26) |
47 (47 - 47) |
30 (21 - 40) |
23 (21 - 26) |
47 (47 - 47) |
0.240 |
Values are median (i.q.r.); # Kruskal Wallis Test, p values<0.05 are considered significant
StO2, Oxygen saturation index; THI, Tissue hemoglobin index; NIR, Near infrared index; TWI, Tissue water index
Point 5: Has any other technique recently been proposed in this setting to evaluate ischemia/tissue perfusion?
Response 5: Thank you for pointing this out. Recently an excellent review concerning viability assessment in liver transplantation and the impact of dynamic organ preservation was published by Panconesi et al. (PMC7915925). The actually available methods/techniques to evaluate viability of the liver during machine perfusion, specifically ischemia/tissue perfusion are:
- Macroscopic assessment
- Ultrasound
- Fibroscan
- Histology
- Haemodynamics during perfusion (vascular resistance, perfusion flows)
- Blood gas analysis
- Biochemical analysis
- Mass spectroscopy and spectrometry
- Metabolomics, proteomics, genomics
We integrated this reference in our manuscript.